# The Antibiotic Treatment of Calf Diarrhea in Four European Countries: A Survey

**DOI:** 10.3390/antibiotics10080910

**Published:** 2021-07-26

**Authors:** Cassandra Eibl, Ricardo Bexiga, Lorenzo Viora, Hugues Guyot, José Félix, Johanna Wilms, Alexander Tichy, Alexandra Hund

**Affiliations:** 1University Clinic for Ruminants, Department for Farm Animals and Veterinary Public Health, University of Veterinary Medicine Vienna, 1210 Vienna, Austria; Cassandra.Eibl@vetmeduni.ac.at; 2Centro de Investigação Interdisciplinar em Sanidade Animal, Faculdade de Medicina Veterinária, Universidade de Lisboa, 1300-477 Lisbon, Portugal; ricardobexiga@fmv.ulisboa.pt (R.B.); jose_duarte_felix@hotmail.com (J.F.); 3School of Veterinary Medicine, College of Medical, Veterinary and Life Sciences, University of Glasgow, Glasgow G61 1GH, UK; Lorenzo.Viora@glasgow.ac.uk; 4Clinical Department of Production Animals, Fundamental and Applied Research for Animals and Health, University of Veterinary Medicine, 1210 Vienna, Austria; Hugues.Guyot@uliege.be; 5Tierarztpraxis Geisenhausen, 84144 Geisenhausen, Germany; johanna.wilms@gmx.de; 6Platform Bioinformatics and Biostatistics, Department for Biomedical Sciences, University of Veterinary Medicine, 1210 Vienna, Austria; Alexander.Tichy@vetmeduni.ac.at; 7Agricultural Center for Cattle, Grassland, Dairy, Game and Fisheries of Baden-Württemberg (LAZBW), 88326 Aulendorf, Germany

**Keywords:** neonatal calf diarrhea, survey, antibiotics, HPCIA

## Abstract

Neonatal calves are commonly affected by diarrhea caused by different pathogens, but not always bacteria. Yet, antibiotics are routinely used as a treatment to an unknown extent. It was our goal to survey antibiotic use for the treatment of neonatal calf diarrhea in different countries and to identify influencing factors. A total of 873 farmers and veterinarians in Austria, Belgium, Portugal, and Scotland participated in a voluntary online survey. The data were analyzed using classification and regression tree analyses and chi^2^ tests. Overall, 52.5% of the participants stated that they use antibiotics when treating neonatal calf diarrhea. Of those, 27% use them always, and 45% use highest priority critically important antibiotics. The most important factor differentiating antibiotic use practices was the country the participants were from, which could be due to regulatory differences between the countries. All antibiotic products stated were licensed for use in cattle, but several were not licensed for the treatment of diarrhea in calves. Our study shows that there is an urgent need for more scientific evidence to define best practices for the treatment of neonatal calf diarrhea. Furthermore, consensual criteria for antibiotic therapy must be defined, and targeted training for farmers and veterinarians must be provided.

## 1. Introduction

### 1.1. Regulatory Basis

Neonatal calf diarrhea (NCD) is the most commonly treated disease in cattle [1,2]. In Europe, the approach to treating sick calves is determined by law to a certain extent: Any calf, which appears to be ill or injured, must be treated appropriately without delay, and veterinary advice must be obtained as soon as possible for any calf that is not responding to the stock keeper’s care [3]. Choosing medical treatment is the responsibility of the attending veterinarian and, depending on the legal situation, the responsibility of the farmer. To which extent farmers can get involved in the treatment of sick animals is regulated at the country level [3,4,5,6]. All antibiotics licensed for use in food-producing animals are prescription-only medicines that may only be administered following a clinical assessment of the animal or group of animals, diagnosis, and prescription by a veterinarian [7]. Ideally and according to best practice, the choice of antibiotic drug is determined by appropriate laboratory tests such as culture and sensitivity testing [8,9]. The veterinarian must weigh the benefits and risks for animals, humans, and the environment based on her or his knowledge and considering the current state of knowledge in veterinary medicine. The veterinarian can then recommend the most appropriate therapeutic treatment by use of the optimal drug, dosage, and duration of treatment [7,10]. Ensuring responsible antibiotic use on-farm is an essential part of a veterinarian’s role, even though they may not be directly administering the medicines [10].

There are no legal regulations governing antibiotic use in detail. However, several national veterinary organizations have developed antibiotic use principles, programs, and algorithms (Table 1). These guidelines are intended to be a practical benchmark for a careful, medically justified use of antibiotics. Both animal and human health could benefit by minimizing the risks associated with the emergence and spread of antimicrobial resistance [11]. Prudent use of antibiotics should lead to more rational and targeted use.

### 1.2. Antibiotic Use in Calves with Diarrhea

There are several issues with the antibiotic treatment of calves with NCD, as the correct indication for treatment and choice of drug is often problematic. The etiological diagnosis is the first important pitfall [12]; Viral and parasitic pathogens are more likely to be involved as primary causes of NCD than bacterial pathogens. Therefore, the majority of antibiotic treatments may not be justified [12,13,14]. 

The decision to administer antibiotics should not be based only on the clinical signs and type of diarrhea or the veterinarian’s clinical experience but on diagnostic testing as well. The detection of *Escherichia coli* (*E. coli*) F5 (K99) or of bacteremia, for example, may warrant the use of an antibiotic. For rapid animal-side testing of fecal pathogens, several point-of-care tests have been described [15,16,17]. A test for the detection of bacteremia in connection with bacteriuria in newborn calves has been validated but is not widely used in practice to date [18].

Aside from *E. coli*, treatment of NCD with oral or injectable antibiotics may only be necessary in cases where the calves show signs of systemic illness such as fever and depression or in calves that have blood or mucosal shreds in their feces, as it marks a breakdown of the blood-gut barrier [19]. The treatment of the concomitant Gram-negative septicemia and bacteremia and the decrease in numbers of coliform bacteria in the proximal small intestine and abomasum is the most important goal of antibiotic therapy in NCD [19,20]. Therefore, the antibiotic must be excreted in bile and reach an effective level in the gastrointestinal tract [21].

Antibiotics may have an impact on the microbiome in the gastrointestinal tract. There are significant differences in microbial diversity between healthy and diarrheic calves within a farm [22,23]. Such microbiome changes in sick calves usually return to the pre-diarrheal stage after a week [24]. It is uncertain if the reduction in microbial diversity occurs due to the disease itself or the antibiotic treatment [22]. A very limited number of studies show that therapeutic antibiotics delay the temporal development of diversity [25]. As an example, the use of tulathromycin for treatment appeared to have a negative impact on the richness and diversity of the gut microbiome [26]. A study in 2009 showed that calves treated with antibiotics or fed with medicated milk replacer had 70% and 31% more days with diarrhea, respectively, compared to calves with NCD that only received antibiotics in cases with fever and depression [27].

The Belgian Knowledge Center for the Use of Antibiotics and Antibiotic Resistance in Animals (AMCRA) does not advise the use of antibiotics as the first-line treatment of NCD. Second choice drugs are sulfonamides with trimethoprim, amoxicillin, amoxicillin and clavulanic acid, colistine, gentamicine, and paromomycin. As the third choice, quinolones and flumequine are recommended, but diagnostic testing (culture and sensitivity) is mandatory beforehand. For septicemia, the drugs of choice are penicillin or sulfonamides with trimethoprim. The second- and third-choice antibiotics for this indication are the same as listed for NCD treatment [28]. In Switzerland, official treatment guidelines for NCD do not recommend antibiotic treatment in simple cases. However, in NCD due to *E. coli* K99, amoxicillin as the first choice and sulfonamides with trimethoprim as the second choice for oral and parenteral treatment are recommended. Neomycin and amoxicillin with clavulanic acid can be used as the third choice for oral treatment. Colistine and quinolones are recommended for restricted use only after culture and sensitivity, and the use of cephalosporins is strictly discouraged due to their low concentrations in the intestinal tract [29].

Outside the EU, Berchtold and Constable (2008) and Constable (2009) propose amoxicillin, ampicillin, and potentiated sulfonamides as first-choice antibiotics for parenteral administration in patients suffering from NCD. For oral administration, amoxicillin or amoxicillin/clavulanate potassium has been recommended. The second choice of antibiotics is third- and fourth-generation cephalosporins, such as ceftiofur and cefquinome. The last-choice antibiotics are fluoroquinolones, which should only be used for the treatment of *E. coli* diarrhea and salmonellosis in calves [19,20,21].

The British Veterinary Association (BVA, London, United Kingdom) has recommended minimal use of third and fourth-generation cephalosporins, fluoroquinolones, and colistin [10]. These drugs should only be used where they have been demonstrated by sensitivity testing to be the only suitable choice to avoid unnecessary suffering.

Unfortunately, even in the absence of known disease, antibiotics are used extensively in calves for both therapeutic and prophylactic purposes worldwide [14,30]. Although selling milk replacer containing antibiotics has been prohibited in the European Union for almost 30 years, it is still common practice in many countries to feed calves prophylactically with medicated milk replacers containing antibiotic agents such as oxytetracycline and neomycin [1,31,32,33,34,35]. In 2012, a Belgian study reported that a reduction in oral antibiotic group treatments for prophylactic and metaphylactic reasons would be the simplest and probably the most efficient way to achieve a reduction in antibiotic use in the veal industry [36].

There is a potential misuse of antibiotics occurring in extra-label use, including with highest priority critically important antimicrobials (HPCIA). These HPCIA contain the antibiotic classes fluoroquinolones, cephalosporins (third and higher generations), macrolides and ketolides, glycopeptides, and polymyxin [37]. Each antibiotic preparation is labeled for certain therapeutic indications. Any deviation and thus extra-label use has to be dictated by a veterinarian and must be justified [10]. It is only allowed in the event of a therapeutic emergency and must not result in violative usage in food-producing animals [38]. According to several international guidelines [10,39,40], extra-label use must be reserved for exceptional circumstances, following appropriate sensitivity testing, and the usage of HPCIA must be restricted for use as a last resort under veterinary direction. However, the extra-label use of antibiotics administered by the farmer and mandated by the veterinarians is reported [1]. Although several antibiotic classes are labeled for treatment of diarrhea in calves [21], extra-label use such as the use of spectinomycin solely or in combination with oxytetracycline in calves is observed quite often as this combination is widely used on farms to prevent diarrhea [1,41,42]. Other recommend antibiotics include ceftiofur hydrochloride for the treatment of diarrhea [31,34]. Macrolides were used in 11% of the cases, where oral antibiotics were administered as treatment [43]. In Sweden, streptomycin is occasionally used to treat diarrhea in calves [44]. Constable et al. (2009) propose that the extra-label use is justified for the treatment of calf diarrhea due to the lack of published studies documenting the clinical efficacy of antibiotics with a label claim for the treatment of calf diarrhea and because of the life-threatening situations that can occur in calves with diarrhea [19]. According to Mohler et al. (2019), most of the drugs effective against Gram-negative bacteria are not labeled for the dose rate that provides therapeutic drug concentrations [45].

It is also reported that calves treated for diarrhea frequently received more than one type of antibiotic agent [14]. Additionally, there is a tendency to rely on personal experience for antibiotic usage and dosage [34,46].

### 1.3. Aim of the Study

There is little information available about decision-making processes concerning the use of antibiotics in treating calves with diarrhea. The aim of our study was to describe the treatment of neonatal calf diarrhea in the four different European countries, Austria, Belgium, Portugal, and Scotland, as part of the United Kingdom, using an online survey. In this part of the study, we focused on specifying factors influencing decision-making in veterinarians and farmers concerning the use of antibiotics. We also compare antibiotic treatment regimens to scientific best practices and national guidelines.

## 2. Results

### 2.1. Respondent Characteristics

A total of 873 questionnaires (Austria: 547, Belgium: 92, Portugal: 163, Scotland: 71) were included in the analysis. Of those, 597 were answered by farmers (female: 138, male: 458, N/A: 1) and 276 by veterinarians (female: 83, male: 192, N/A: 1). Based on the results of Vetsurvey 2018 (total numbers of veterinarians in each country), 17.6% of the veterinarians in Austria, 1.4% in Belgium, 2.6% in Portugal, and 0.3% in the United Kingdom participated in our study [47]. The age of the participants ranged from 18 to 75 years (*n* = 870, 32.9 ± 13.3; 30 years; mean ± SD; median). Most of the participants were Austrian farmers (*n* = 446) with a median age of 25 years. In terms of experience, 23.1% of the participants were working with both dairy and beef cattle (called mixed in the following text), 55.5% were mainly working with dairy cattle, and 21.2% with beef cattle only (*n* = 872).

### 2.2. Use of Antibiotics for the Treatment of NCD

As shown in Figure 1, 458 participants out of 873 stated that they used antibiotics in calves suffering from NCD. Country was the most important variable (normalized weight 100%) for the differentiation of antibiotic use in calves with NCD, followed by occupation and age (91.1% and 46.1%). Experience only accounted for 10.9% and sex for 0.3% in normalized weight. Austrian veterinarians and farmers used significantly fewer antibiotics than participants from the other countries. Based on occupation, Austrian farmers administered fewer antibiotics compared to veterinarians.

Logistic regression analysis showed a significant impact of age on the probability of antibiotic use for the treatment of NCD in veterinarians (Figure 2, intercept = 2.45; slope = −0.026; *p* = 0.031). The younger the veterinarians were, the higher the probability of using antibiotics. However, even in older veterinarians, the probability was still over 60%. Regarding the use of HPCIA, there was no significant relation with age in veterinarians.

### 2.3. Situations Where Antibiotics Are Being Used

Of the 458 respondents using antibiotics for the treatment of NCD, 404 participants provided more information in the question “if you usually use antibiotics-please state when” and “please state approximate %”. Of those, 30.7% (*n* = 124) stated that they used antibiotics always and 69.3% (*n* = 277) in some situations (Appendix A), namely in an average of 48.5% ± 28.7% (median 50%). CART analysis resulted in the country as the most important factor to classify the frequency of antibiotic use, followed by experience with normalized importance of 68.6%, age (42.3%), occupation (7.1%), and sex (4.5%). Participants from Portugal and Scotland used antibiotics significantly more frequently in every case without differentiating further (in some situations: 54.1%, *n* = 92; always: 45.9%, *n* = 78) compared to respondents from Austria and Belgium (in some situations: 80.3%, *n* = 188; always: 19.7%, *n* = 46). Out of the total of 88 respondents working with beef cattle in Scotland and Portugal, 65 (farmers and veterinarians) used antibiotics for the treatment of NCD, and 44% of those (*n* = 39) stated that they used them always when treating NCD. Participants younger than 41.5 years working with dairy cattle and mixed cattle used antibiotics significantly more often in every case of NCD compared to older participants (>41.5 years).

A total of 227 participants specified the situations when they used antibiotics in treating NCD. Most of the participants (*n* = 183) said that they used them in calves with NCD when their body temperatures were above normal (>39.5 °C) or when they had blood in the feces (hematochezia, *n* = 164). Calves suffering from NCD that had very watery diarrhea or were not able to stand were treated with antibiotics by 157 and 137 participants, respectively. Absence of suckling reflex (*n* = 101), sunken eyes (*n* = 82), cold mouth (*n* = 79), and an internal temperature below normal (hypothermia, <38.0 °C, *n* = 65) were used less frequently as indication to use antibiotics.

The question “specify the situation: others” was a free text answer and was answered by 22 participants. They stated that they used antibiotics in the following situations: dehydration, mucosal shreds in the feces or signs of sepsis (e.g., increased episcleral vascular injection), if homeopathy does not help, when the duration of diarrhea is longer than two days, another organ is affected (e.g., bronchitis), a negative result of rota- and coronavirus rapid test, based on the appearance of the stool, if *E. coli* or *Salmonella* infection are suspected and one participant stated that this depended on the calf’s age.

Pearson chi-square test revealed that, compared to veterinarians, farmers administered significantly more often antibiotics when calves had watery feces (*p* = 0.016), whereas veterinarians chose to administer antibiotics when the calves were not suckling (*p* = 0.014), had sunken eyes (*p* = 0.001), a body temperature above or below normal (*p* < 0.001 and *p* = 0.017, respectively) and blood in the feces (*p* = 0.001). Based on CART analysis, body temperature above normal was the most important factor comparing veterinarians and farmers: 114 of 132 (86.3%) veterinarians and 69 of 154 farmers (44.8%) said that they used antibiotics in this case.

Women stated significantly more often that they would use antibiotics when body temperature was above normal compared to men (*p* = 0.007). Participants working with beef cattle administered significantly more antibiotics compared to participants working with dairy cattle or in the mixed sector (*p* < 0.0001); regarding the use of HPCIA, there was no difference. Participants working with dairy used significantly fewer antibiotics in calves with a body temperature above normal (*p* = 0.003).

### 2.4. Antibiotic Classes as First, Second, or Third Choice

When farmers and veterinarians were asked, “Which antibiotic do you use as first, second, and third choice for the treatment of calves with diarrhea?”, quinolones, sulfonamides, and penicillins were the three antibiotic classes most frequently named in all three choice categories (Figure 3). All antibiotics that were stated in this questionnaire were licensed for use in cattle, but several were not licensed for the treatment of diarrhea in calves specifically, including ceftiofur and cefoperazon (third generation cephalosporins), tulathromycin and tilmicosin (macrolids), as well as florfenicol (fenicoles). All used quinolones were licensed for cattle and most of them for treatment of infections of the gastrointestinal tract caused by enrofloxacin, danofloxacin, or flumequin susceptible strains of *E. coli* (e.g., Advocid^®^, Enrosleecol^®^, Fluyesyva inyectable^®^). However, several marbofloxacin drugs (e.g., Marbocyl^®^, Marbosyva^®^, Marbox^®^, Ubiflox^®^) were stated as well, although they were only licensed for the treatment of mastitis and respiratory infections.

Some participants stated registered trade names that did not contain antibiotics but could be used for treating calves with NCD, for example, NSAIDs (Tolfedine^®^), parasympatholytics (Buscopan^®^), oral rehydration solutions (Elektrydal^®^, Nutrivet total^®^) and antiparasitic drugs (Baycox^®^, Halocur^®^).

### 2.5. Use of Oral and Injectable Antibiotics

In questions 6 and 7, participants were asked if they used oral and/or injectable antibiotics as treatment in calves with diarrhea. The highest proportions of respondents stated they would use injectable antibiotics (Table 2).

### 2.6. Use of HPCIA for Treatment of NCD

The majority (206 out of 291) of participants who answered questions on the type of antibiotic they were giving to calves with NCD named at least one class of HPCIA as their choice (Figure 4). Again, country was by far the most important factor for differentiating the use of HPCIA. In relation to country, the normalized importance of sex, age, experience, and occupation accounted for only 27.6%, 21.0%, 20.5%, and 16.8%, respectively. Veterinarians and farmers in Scotland named significantly fewer brands of HPCIA drugs compared to participants in Austria, Belgium, and Portugal. Based on sex, Scottish women used HPCIA significantly more often than men.

### 2.7. Use of HPCIA According to Situation

Almost 70% of the participants named at least one HPCIA as the drug of choice for the treatment of NCD when they also chose “calves had a body temperature above normal” as the reason for antibiotic treatment. Approximately 60% named HPCIA and chose calves suffering from watery diarrhea as reason, whereas 50% named HPCIA as the drug of choice and chose calves that were not standing or had bloody feces as a reason to administer antibiotics. Based on CART results, “body temperature above normal” received the highest importance for the decision to administer HPCIA, followed by watery feces (19.1%) and sunken eyes (11.1%). The absence of a suckling reflex (84.3%), blood in the feces (65.2%), body temperature below normal (54.9%), and cold mouth (16.4%) were chosen as a reason for antibiotic treatment by more participants who did not name an HPCIA as the drug of choice.

## 3. Discussion

This survey was carried out to gain insight into the factors that would influence the decision of veterinarians and farmers in four different European countries on whether or not to use antibiotics for the treatment of NCD. The survey was conducted with veterinarians and farmers volunteering to answer the questionnaire, thus yielding different amounts of participants from each country.

Over 50% of respondents of the survey stated that they used antibiotics in calves affected by NCD. Some people even stated that they use them always, and others used them about half of the time. The use of antibiotics to treat NCD might not be necessary: most cases of NCD are caused by viral and parasitic pathogens, with *E. coli* K99 being the third most prevalent infectious agent in NCD worldwide [48,49,50]. Therefore, blanket antibiotic treatment of NCD should be strongly discouraged.

Aside from bacterial pathogens as a cause of NCD, calves that are affected by septicemia as a result of NCD must be treated using antibiotics. Studies have shown that severely ill calves may be bacteremic and, as a result, septic, especially if they are very young (<5 days) and affected by the failure of passive transfer of immunity [50,51,52,53]. Bacteremia cannot be diagnosed based on clinical signs; it may be detectable if it occurs in connection with constant bacteriuria using a catalase-based calf-side urine test [18]. However, septicemia, the systemic inflammatory response (SIRS), can be diagnosed by performing a thorough clinical examination [53,54,55]. Trefz et al. (2016) based clinical evidence of septicemia on marked hyperaemia of mucous membranes, congestion/injection of episcleral vessels, mucosal or subscleral bleeding, or hypopyon. The authors assumed the presence of SIRS in calves with two of the clinical criteria hyperthermia or hypothermia (reference interval, 38.5–39.5 °C), tachycardia (>120 beats/min), and tachypnoea (>36 breaths/min) [56]. Constable (2004) stated that potential *E. coli* bacteremia should be treated in calves with diarrhea that have a reduced suckle reflex, marked dehydration, weakness, inability to stand, or clinical depression [21]. Most participants of our study specified fever and hematochezia as an indication to administer antibiotics to calves suffering from NCD. Both could be signs of sepsis and the disruption of the blood-gut barrier and are therefore reasonable choices [13,53,54]. The same two clinical signs, fever and hematochezia, were used to develop a simple algorithm for the treatment of calves affected by NCD, which lead to a significant reduction in antibiotic use with no changes in morbidity and mortality [13].

Interestingly, hypothermia, or cold mouth as a sign of it, was chosen by the fewest participants as reasons for antibiotic treatment [13]. Of course, those signs, such as sunken eyes and very watery diarrhea, could be related to dehydration and do not necessarily warrant the use of antibiotics, along with the inability to stand or suckle, which could be due to D-lactic acidosis [57]. Some participants even pointed out that factors such as the duration of sickness or color of feces play a significant role as well. Such findings show the urgent need for implementing an algorithm for treating NCD and restricting antibiotic use to calves with defined clinical symptoms.

Using the above-mentioned clinical signs as guidance for treatment decisions in calves with NCD together with commercially available rapid /point-of-care tests would be a valuable contribution to the reduction in antibiotic use in calves [15,16,17]. In the authors’ opinion, there is a real need for well-constructed intervention studies testing treatments alongside recorded clinical signs and secondary factors. Additionally, the impact of antibiotic treatment and the consequences for the gastrointestinal microbiome need to be characterized further.

To the authors’ knowledge, there are no studies investigating clinical signs in calves with NCD and benefits from antibiotic treatment. It has been observed that veterinarians use several factors to make the decision to use a certain antibiotic drug. Clinical factors such as clinical signs, expected pathogen, the spectrum of activity of the drug, experience using the drug on own farm, response to previous therapy, ease of administration, the farmer’s ability to administer the drug, recommended frequency of treatment, and drugs cost were their main reasons for choosing an antibiotic drug over another [58,59,60].

However, non-clinical drivers such as the veterinarian-farmer relationship were just as relevant [58,61,62] and could play a further role in the increased use of antibiotics by veterinarians. Fear of unsuccessful treatment, lack of confidence in the diagnosis, and dairy farmer’s demands are the major influencers for veterinarians’ antibiotic use even in conditions not requiring antibiotic use. An increased workload can also play an important role, as veterinarians fear to revisit when the animal did not improve after the first treatment, and they are called again [58]. Furthermore, veterinarians fear they will be blamed if antibiotics later prove necessary [58,63]. Similar results are reported in a Dutch study in 2015, where veterinarians confirmed that the perceived pressure from clients can be a driver for antibiotic use [61].

Farmers from Austria, Belgium, and Portugal administered significantly fewer antibiotics and HPCIA compared to veterinarians. This could be due to legislation in Austria, with the largest proportion of farmers participating, where veterinarians must see an animal before prescribing antibiotics for this specific animal only. Therefore, it would be common for farmers not to have antibiotics in stock because they would not be allowed to make treatment decisions without a veterinarian. In a study from New Zealand, farmers stated that the most important factor (besides the veterinarian’s advice) for the selection of antibiotics was their own experience [59]. Meanwhile, apart from themselves, veterinarians identified farmers as the people having the most important role in responsible antibiotic use, especially as it is often farmers who make the treatment decisions on-farm. Therefore, it is necessary to make sure they are sufficiently informed about the etiology of calf diarrhea and the use of HPCIA, and they understand antibiotics classes, indications, and dosages [43]. However, some farmers in our study named drugs such as butylscopolamine bromide and metamizole, as well as oral electrolytes as antibiotic agents in questions 19 to 21. This may occur because some farmers do not know the pharmacological properties of drugs they use for treating sick calves or because they misunderstood the question. According to Sawant et al. (2005), the main reasons for farmer’s misuse of antibiotics on-farm are failure to consult a veterinarian for treating sick animals, absence of antibiotic treatment records, and lack of written protocols for treating sick animals [1]. Such often simple and cost-effective treatment protocols or algorithms for antibiotic selection for diarrhoeic calves have been proved successful as means of a reduction in antibiotics usage as these guidelines lead to a more rational use [13,27].

In several European countries, national and international bodies have developed and issued a variety of recommendations and treatment guidelines in recent years to reduce inappropriate prescribing and antibiotic use [7,10,11]. However, there is no widely agreed simple guide for farmers and vets for NCD treatment to help reduce antimicrobial resistance. An example of such a guide can be found in the Teagasc Calf Rearing Manual [64].

The effectiveness of such guidelines is questionable: although 90% of bovine veterinarians stated that they read cattle-related journals regularly, official reports were considered less popular information sources [65]. Instead, practitioners said they value training/literature, experience, label, sensitivity testing results, and universities as the most important information sources, which influenced their antibiotic prescribing behaviors. Almost 80% of the veterinarians frequently participated in cattle medicine-trainings such as meetings, workshops, and congresses [65]. Therefore, continuing veterinary education of veterinarians, who are the first line of information to farmers, is a key to reducing antibiotic use and, particularly, those HPCIA [13].

Besides country, age was the second most important factor regarding normalized weight. The median age of 30 years of all participants in our study may be because younger people were more likely to fill out an online survey than older people, who prefer paper and pencil surveys [66]. We suspect that older farmers asked their more technology-experienced children for help filling in the online questionnaire, who then stated their own data (e.g., age, sex).

The probability of using an antibiotic for the treatment of NCD decreased with increasing age in veterinarians. Krupat et al. (2000) found out that human patients were more satisfied with physicians whose orientation was congruent with theirs than those who had a different opinion [67]. However, older and more experienced physicians were better able to refuse patients’ demands [68,69]. Although those studies apply to human medicine, it is reasonable to assume that veterinarians are subject to the same mechanisms in the veterinarian-client interaction and that older veterinarians are more likely to follow treatment plans that they consider most appropriate.

One of the most important factors in veterinarians governing the selection of an antibiotic for treatment is their own experience [46,59,63]. The lack of experience and confidence might be a reason for a higher amount of antibiotics used for the treatment of NCD carried out by younger colleagues. A small survey including staff of a veterinary teaching hospital in the U.S. showed that veterinarians who graduated after 1999 were less concerned about antibiotic resistance and judicious use of antibiotics than older colleagues [60]. Such an attitude could reflect an inadequate emphasis on training of our younger graduates in some schools more focused on small animal cases [60,70].

Fluoroquinolones, sulfonamides, and penicillins were the most frequently specified classes for the antibiotic treatment of NCD in our study. This outcome is similar to previous studies [31,34], including an Italian survey, where quinolones were quoted by 54% of the surveyed veterinarians as their first choice and by 38% as their second choice for the treatment of diarrhea in calves [65]. A study carried out in Switzerland showed that the common treatment of calf diarrhea consisted of fluoroquinolones, which were used in 47% of the parenteral treatments [43]. These results indicate that there is a variety of antibiotics and HPCIA that are used for the treatment of NCD, despite questionable efficacy [57]. Antibiotics should only be used in calves suffering from NCD that are also affected by Gram-negative septicemia, mainly caused by *E. coli* [19,21]. As mentioned previously, studies differentiating infectious agents responsible for NCD show that *E. coli* only affects a small proportion of calves, making it only the third most prevalent cause of NCD worldwide [48,49]. Aggravatingly, as already mentioned, there is a lack of well design studies to determine the most effective antibiotic treatment for Gram-negative septicemia in calves suffering from NCD.

Several antibiotics that are not licensed for use in NCD treatment were cited from veterinarians and farmers as first, second, or third choice in this study. The reason for this is probably be found in the lack of knowledge on licensed indications. This leads to extra-label use and treatment decisions, which are based on beliefs of efficacy rather than science [71]. Many practices adopted in the field are not evidence-based. A reason for the frequent choice of quinolones as the second or third choice may be based on the fear of septicemia when a non-HPCIA does not work as a drug of first choice.

Veterinarians from Scotland stated less HPCIA as drugs of choice in calves with NCD. This could be due to a better understanding of the prudent use of antibiotics, as implemented in the Red Tractor program [72]. Herein, among other measures, the use of HPCIA is only allowed as a last resort under veterinary direction, backed up by sensitivity or diagnostic testing. Antibiotic failures must be discussed. Staff, which is responsible for medicine administration, is instructed to attend training courses (handling, correct administration storage conditions, purchasing routes). In the case of non-conformance, including repeated use of HPCIA without testing, there is an impact on certification.

Our study shows clearly that similar programs need to be implemented in all European countries to increase the awareness of prudent antibiotic use in the farming and veterinary community.

## 4. Materials and Methods

### 4.1. Questionnaire

A questionnaire (http://biosegur.fmv.ulisboa.pt/index.php/356164/lang-en) (access date 25 June 2021) was designed by RB for collecting information on the treatment of neonatal calf diarrhea (NCD). The questionnaire was translated and made available to veterinarians and dairy and beef farmers in Portugal (RB), Belgium (HG), Scotland (LV), Austria, and adjacent German-speaking countries (summarized to Austria in the following text; AH). The surveys were conducted for a limited number of weeks per country from February 2016 until January 2019. The veterinarians and farmers were informed via newsletters and through e-mails from various organizations (e.g., vet board) or during conferences. The survey was available online, and some questionnaires were filled out during farm visits or by veterinarians at a conference or over the phone. The entire questionnaire covered many aspects of medical treatment of NCD and husbandry practices regarding sick calves. The part applying to antibiotic treatment consisted of a maximum of 21 questions (Table 2). The questionnaire included ‘yes’ or ‘no’ questions, single- and multiple-choice questions, as well as open-ended questions. For questions 19 to 21 (first-, second-, and third-choice antibiotics), commercial names or drug names were accepted as possible answers. To decrease reactance, the forced-choice answer format was avoided; therefore, the number of responses per question varies.

All questionnaires were individually examined for aberrant results and plausibility before statistical analyses. In order for questionnaires to be included in the study, 2 out of 4 personal questions (Table 3 question 1–4) and at least one technical question had to be answered.

### 4.2. Data Analysis

All statistical analyses were performed using IBM SPSS v24. Differences in frequency distributions were analyzed using Pearson’s chi-square test. Logistic regression analysis was performed to model the impact of age on the probability of antibiotic use. Classification and regression tree (CART) analyses were carried out to predict the use of antibiotics, HPCIA, and the frequency of antibiotics use based on the given information’s about participants (country, occupation, experience, sex, and age) or signs (e.g., bloody feces, sucking reflex, temperature above or below normal). Every factor that is added to the model receives a value for its importance within the classification process. The importance is calculated using the GINI-Index. As a result, the importance of each factor is given in percentages in relation to the most important factor (normalized importance). Trees were pruned to avoid too complex trees. As a stopping rule, the minimal size for parent nodes was set to 25, the minimal size for child nodes was set to 10. For all analyses a *p*-value below 5% (*p* < 0.05) was seen as significant.

## 5. Conclusions

This study illustrates that there may be excessive use of antibiotics and HPCIA for the treatment of NCD. The younger the veterinarians were, the higher the probability of using antibiotics. Even in older veterinarians, the probability of using antibiotics was still over 60%. Most respondents stated that they would choose to administer antibiotics in calves with fever and bloody feces, which could be indicators for sepsis and indeed warrant antibiotic use. However, it is very likely that antibiotic use could be substantially decreased in the treatment of calves with NCD implementing specific guidelines and targeted training for veterinarians and farmers. Even without better scientific evidence, it is clear that many veterinarians and associated farmers are not applying best practice and agreed overall guidance similar to that seen in SCOPs (Sustainable Control of Parasites in Sheep) for anthelmintics in the U.K. is sorely needed.

## Figures and Tables

**Figure 1 antibiotics-10-00910-f001:**
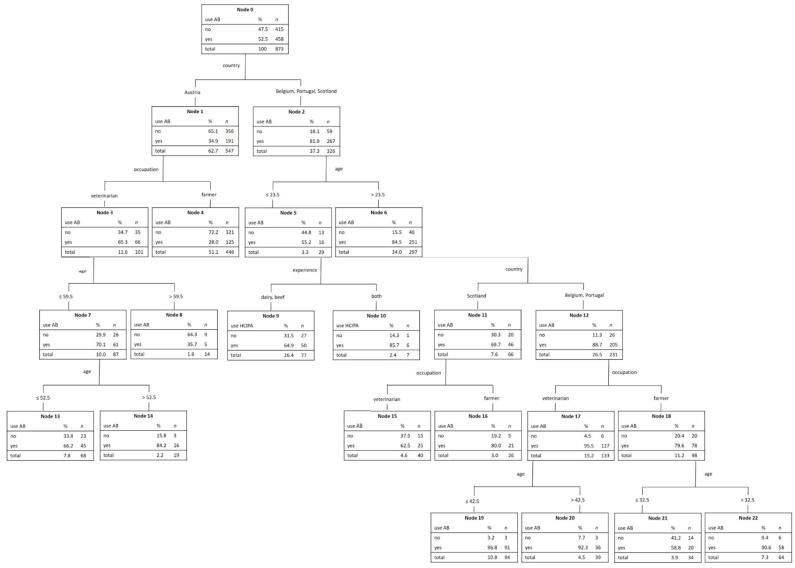
CART: Association of different factors on the use of antibiotics for the treatment of NCD.

**Figure 2 antibiotics-10-00910-f002:**
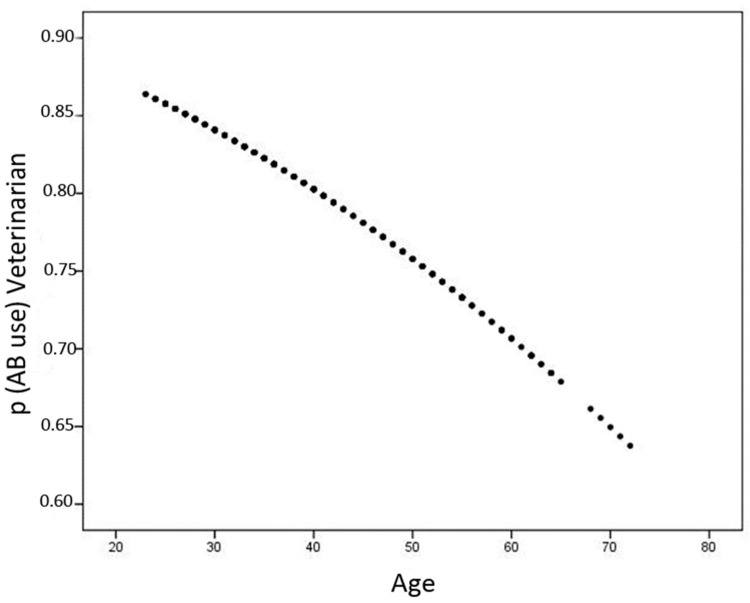
The probability of veterinarians using an antibiotic for the treatment of NCD depended on their age.

**Figure 3 antibiotics-10-00910-f003:**
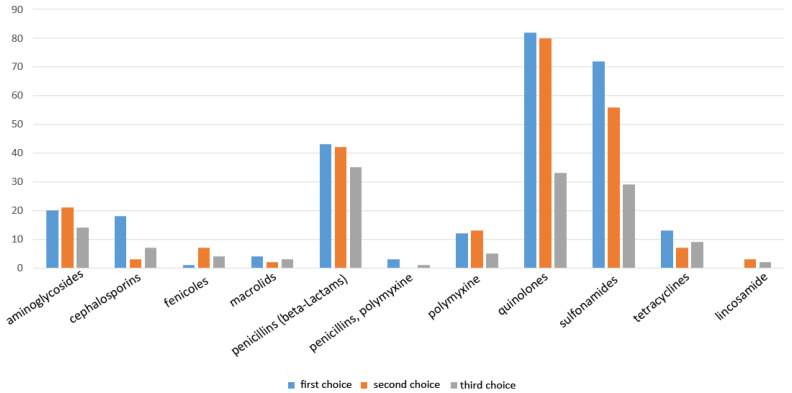
Use of antibiotic classes as first, second, or third choice.

**Figure 4 antibiotics-10-00910-f004:**
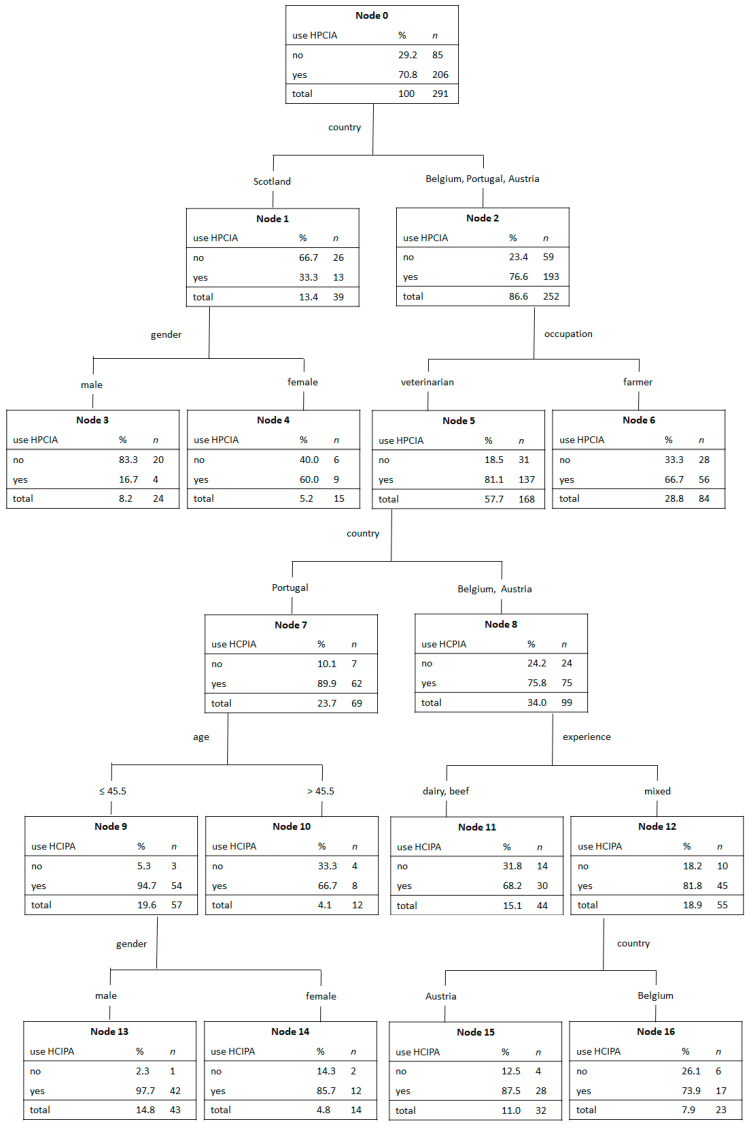
CART: Association of different factors on the use of HPCIA for the treatment of NCD.

**Table 1 antibiotics-10-00910-t001:** Summary of guidelines for antibiotic use.

Country	Guideline
Austria	Leitlinien für den sorgfältigen Umgang mit antibakteriell wirksamen Tierarzneimitteln des Bundesministeriums für ASGK (BMASGK-74330/0008-IX/B/15/2018, AVN Nr. 2018/11a) Umgang mit antibakteriell wirksamen Tierarzneimitteln- Leitfaden für die tierärztliche Praxis, Bundesministeriums für ASGK und Österreichische Tierärztekammer 2019
Belgium	AMCRA- Kenniscentrum inzake antibioticagebruik en -resistentie bij dieren: Richtlijnen voor goed Gebruik van antibiotica, June 2016 Royal Decree, July 2016 (conditions of use of drugs for veterinarians and farmers)
Portugal	No such guidelines have been published by official or professional bodies
United Kingdom	British Veterinary Association: BVA policy position on the responsible use of antimicrobials in food-producing animals, May 2019 British Veterinary Association: Responsibly use of antimicrobials in veterinary practice: the 7-point plan, 2019 British Cattle Veterinary Association: AMR Statement, December 2016 RUMA (Responsible use of medicines in agriculture alliance) guidelines for farmers and veterinarians: Responsible use of antimicrobials in cattle production, May 2015
International	EU: Guidelines for the prudent use of antimicrobials in veterinary medicine (2015/C 299/04) WHO guidelines on use of medically important antimicrobials in food-producing animals 2017 WHO list of Critically Important Antimicrobials for Human Medicine (WHO CIA list) 2017

**Table 2 antibiotics-10-00910-t002:** Number of participants using oral and injectable antibiotics by occupation and experience.

Professional Group	Oral Antibiotic	Injectable Antibiotic	Total Answers
Beef veterinarians	21	56	67
Dairy veterinarians	36	67	98
Mixed veterinarians	43	76	110
Beef farmer	18	54	118
Dairy farmer	47	116	387
Mixed farmer	17	18	91

**Table 3 antibiotics-10-00910-t003:** Questions of the survey relevant for the use of antibiotics in treating NCD.

Number	Question	Answer Options
1	Country	Individual answer
2	Profession	Veterinarian/farmer
3	Age	Individual answer
4	Sex	Female/male
5	Type of animal you have more experience with	Dairy/beef
6	In your approach to calves with diarrhea, do you usually use as treatment: oral antibiotic	Yes/no
7	In your approach to calves with diarrhea, do you usually use as treatment: injectable antibiotic	Yes/no
8	If you usually use antibiotics, please state when	Individual answer
9	Please state approximate %	Individual answer
10	Specify the situations: calf is not standing	Yes/no
11	Specify the situations: calf has no sucking reflex	Yes/no
12	Specify the situations: calf has sunken eyes	Yes/no
13	Specify the situations: calf has watery diarrhea	Yes/no
14	Specify the situations: calf has rectal temperature below normal (<38.0 °C)	Yes/no
15	Specify the situations: cold mouth/cold extremities	Yes/no
16	Specify the situations: calf has rectal temperature above normal (>39.5 °C)	Yes/no
17	Specify the situations: calf has blood in the faces	Yes/no
18	Specify the situations: other	Individual answer
19	What are the brand names of the antibiotics you most frequently use: 1st choice	Individual answer
20	What are the brand names of the antibiotics you most frequently use: 2nd option	Individual answer
21	What are the brand names of the antibiotics you most frequently use: 3rd option	Individual answer

## Data Availability

The data presented in this study are available upon request.

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
