# Peer review of "The Antibiotic Treatment of Calf Diarrhea in Four European Countries: A Survey"

_antibiotics, 2021, doi:10.3390/antibiotics10080910_

Round 1

Reviewer 1 Report

The antibiotic treatment of calf diarrhea in four European countries: a survey

The objective of this study was to determine the antibiotic use practices for neonatal calf diarrhea in four countries and the decision-making processes concerning use.  This is an important topic because overuse and misuse of antibiotics can lead to AMR.  NCD is often not caused by bacterial pathogens, and therefore the use of antibiotics is not warranted.  The authors used questionnaires distributed to farmers and veterinarians to obtain the data.

Abstract – the final sentence needs to be restructured. 

Materials and Methods

Questionnaires – how were farmers and veterinarians made aware of the online questionnaire?  Did the authors take any steps to try to ensure equal representation among countries and among the different veterinarians/farmers?

Are the authors able to compare the number of respondents to the number of veterinarians/farmers in the different countries?  Do you have any idea about geographic representation?  I know these numbers may be difficult to figure out but it would be nice to put the number of respondents into a broader context for each country. 

Results

Lines 57-60 – This sentence is clear when you look at the figure but I wasn’t sure reading the sentence if you meant that quinolones were first, sulfonamides were second, etc.  Maybe clarify that all three antibiotics were most frequently named in all three choice categories.

Discussion

Lines 116-135 – This paragraph needs to be re-written to better describe to the readers why the authors are describing these previous studies.  The authors present several different studies without tying them back to their data.  This paragraph would be much more impactful if they told a story with these references and tied the references in to their data.

Lines 151-159 – how do these studies compare to the data differences the authors saw between antimicrobial use decisions by farmers versus veterinarians.

Lines 214-216 – this is really surprising to me.  I would think that current veterinary medicine training would place a larger emphasis on AMR education.

Author Response

Reviewers' Comments to Author and answers:

To the reviewers,

Thank you both for your very thorough work with our manuscript. We have tried to answer all comments on our best behalf. We submitted a copy with tracked changes.

kind regards

Reviewer 1

The objective of this study was to determine the antibiotic use practices for neonatal calf diarrhea in four countries and the decision-making processes concerning use. This is an important topic because overuse and misuse of antibiotics can lead to AMR. NCD is often not caused by bacterial pathogens, and therefore the use of antibiotics is not warranted. The authors used questionnaires distributed to farmers and veterinarians to obtain the data.

Abstract – the final sentence needs to be restructured.

  • Line 29-32: Changed as requested: “Our study shows that there is an urgent need for more scientific evidence to define best practice for treatment of neonatal calf diarrhea. Furthermore, consensual criteria for antibiotic therapy must be defined. Based on that, targeted training of farmers and veterinarians must be provided.”

Materials and Methods

Questionnaires – how were farmers and veterinarians made aware of the online questionnaire? Did the authors take any steps to try to ensure equal representation among countries and among the different veterinarians/farmers?

  • Line 266-267: We added the following sentence: “The veterinarians and farmers were informed via newsletters, and through e-mails from various organizations (e.g. vet board) or during conferences.”
  • Unfortunately, we were not able to ensure equal representation in any way, so our study is based on a convenience sample.

Are the authors able to compare the number of respondents to the number of veterinarians/farmers in the different countries? Do you have any idea about geographic representation? I know these numbers may be difficult to figure out but it would be nice to put the number of respondents into a broader context for each country.

  • We did not record any information in the questionnaire regarding the address of the respondents, so we do not have any information on the location of our respondents. Most likely, the respondents had some geographical proximity to the vet school in their country that conducted the questionnaire.
  • Line 167-170: Added as requested: “Based on results of Vetsurvey 2018 (total numbers of veterinarians in each country) [47], 17.6 % of the veterinarians in Austria, 1.4 % in Belgium, 2.6 % in Portugal and 0.3 % in the United Kingdom participated in our study.”

Results

Lines 57-60 – This sentence is clear when you look at the figure but I wasn’t sure reading the sentence if you meant that quinolones were first, sulfonamides were second, etc.  Maybe clarify that all three antibiotics were most frequently named in all three choice categories.

  • Line 57-60: Changed as requested: When farmers and veterinarians were asked “Which antibiotic do you use as first, second and third choice for the treatment of calves with diarrhea” quinolones, sulfonamides and penicillins were the three antibiotic classes most frequently named in all three choice categories (Figure 3).”

Discussion

Lines 116-135 – This paragraph needs to be re-written to better describe to the readers why the authors are describing these previous studies.  The authors present several different studies without tying them back to their data.  This paragraph would be much more impactful if they told a story with these references and tied the references in to their data.

  • Line 116-148: We changed the paragraph as follows: “Aside from bacterial pathogens as cause of NCD, calves that are affected by septicemia as a result of NCD must be treated using antibiotics. Studies have shown that severely ill calves may be bacteremic and as a result septic, especially if they are very young (< 5 days) and affected by failure of passive transfer of immunity [51-53]. Bacteremia cannot be diagnosed based on clinical signs; it may be detectable if it occurs in connection with constant bacteriuria using a catalase-based calf-side urine test [18]. But septicemia - the systemic inflammatory response (SIRS) – can be diagnosed by performing a thorough clinical examination [53-55]. Trefz et al (2016) based clinical evidence of septicaemia on marked hyperaemia of mucous membranes, congestion/injection of episcleral vessels, mucosal or subscleral bleeding, or hypopyon. The authors assumed the presence of SIRS in calves with two of the clinical criteria hyperthermia or hypothermia (reference interval, 38.5–39.5 °C), tachycardia (> 120 beats/min), and tachypnoea (> 36 breaths/min) [56]. Constable (2004) stated that potential E. coli bacteremia should be treated in calves with diarrhea that have a reduced suckle reflex, marked dehydration, weakness, inability to stand, or clinical depression [21]. Most participants of our study specified fever and hematochezia as an indication to give antibiotics to calves suffering from NCD. Both could be signs of sepsis and the disruption of the blood-gut barrier and are therefore reasonable choices [13,53,54]. The same two clinical signs, fever and hematochezia, were used to develop a simple algorithm for the treatment of calves affected by NCD, which lead to a significant reduction in antibiotic use with no changes in morbidity and mortality [13].

Interestingly, hypothermia, or cold mouth as a sign of it, were chosen by the fewest participants as reasons for antibiotic treatment [13]. Of course, those signs, like sunken eyes and very watery diarrhea, could be related to dehydration and do not necessarily warrant the use of antibiotics, along with the inability to stand or suckle, which could be due to D-lactic acidosis [57]. Some participant even pointed out, that factors like the duration of sickness or color of feces play a significant role as well. Such findings show the urgent need for implementing algorithm for treating NCD and restricting antibiotic use to calves with defined clinical symptoms.

Using the above-mentioned clinical signs as guidance for treatment decisions in calves with NCD together with commercially available rapid /point-of-care tests would be a valuable contribution to the reduction of antibiotic use in calves [15,16,17]. In the authors’ opinion there is a real need for well constructed intervention studies testing treatments alongside recorded clinical signs and other factors.”

Lines 151-159 – how do these studies compare to the data differences the authors saw between antimicrobial use decisions by farmers versus veterinarians.

  • Results of these studies only represent data based on veterinarians. Therefore, we cannot compare farmers and veterinarians antibiotic use based on these studies. We hypothesize that the difference may be caused by the large number of Austrian farmers that do not decide to give antibiotics by themselves because the vet has to make the initial treatment decision and dispense drugs for follow-up treatment. This is explained in the lines 166 – 171 (Discussion).

Lines 214-216 – this is really surprising to me. I would think that current veterinary medicine training would place a larger emphasis on AMR education.

  • It is surprising to us as well. This is a small survey from a veterinary teaching hospital in the US. This could explain the finding. We added the information to the text (lines: 220-221).

Thank you very much for the comments and feedback and the effort you put into our work.

Reviewer 2 Report

Antibiotics-1293869

The indiscriminate use of antimicrobials to treat calves manifesting diarrhea has been a global concern. However, there are few surveys measuring the degree of responsibility of producers and technicians. The aim of this study was to describe the antibiotic treatment of neonatal calf diarrhea in the four different European countries Austria, Belgium, Portugal and Scotland as part of the United Kingdom using an online survey. The results presented showed that most participants have used antimicrobials to treat diarrhea, and there is some evidence of excessive use of antibiotics and HPCIA for the treatment of NCD. This kind of data is important to highlight the subject, and create demand for specific guidelines and targeted training for veterinarians and farmers.

Despite the highlighted importance of this research, there are several major issues to be addressed by the authors.

Major points:

The authors indicate the restricted use of antimicrobials only for bacterial gastrointestinal infections, however there are limited on-farm tests to detect diarrhea pathogens, besides of high costs. Moreover, the dysbiosis process in the gut, independently of pathogens, can justify the use of antimicrobials in some clinical situations, as recommended by Constable. So, I think the authors should explain this consequence of diarrhea in their manuscript. In my clinical experience, most calves with Crypto diarrhea develop septicemia due to intestinal dysbiosis. Microbiome studies have shown that diarrhea can be a consequence or cause of gut microbiome dysbiosis. So, in my opinion the clinical examination is more important than the identification of infectious agents in the fecal samples. It can help, however, a lot of clinical evidence should be considered, as the authors mentioned in the L116-L135. I recommended an introduction and discussion revision explaining these considerations.

Only a 1/4 % of participants answered the specific part of the questionnaire that specified which factors influencing decision-making concerning the use of antibiotics. So, the most of participants maybe did not is the person responsible to make the antimicrobial decision in the farm. What is the author interpretation for this finding? please, include this discussion in the text

It is very nice to present the regulatory rules to the use of antimicrobials in Europe, however what is the experience of authors regarding the effectiveness of penicillin? sulfa and trimetropin ? amoxicillin ? to treat diarrhea. In my experience the effectiveness is very poor. Show me literature and data regarding this indication. In real life, the indication presented in this paper is effectiveness? 

There are some indications to use Ceftiofur to treat calves manifesting diarrhea and sepsis symptoms. See : textbook of the diseases of cattle, horses, sheep, pigs and goats - two-volume set. Authors: Peter Constable et al.

How to balance the regulatory instruction with the in vivo treatment response?

Minor points:

L17-L77 - Add biliary excretion to achieve the intestinal lumen

L88 due to E.coli K99

Add the questionnaire as a supplement document. The link in the M & M is not enough, because is necessary to fill it to follow for the next page

Review the numeration of tables and figures

Tables and figures is not self-explanatory

I did not found figure A1 (cited at L16 page 6)

Table 2 page 8 - absolute numbers or frequencies? it could be performed chi-square

Thanks for the opportunity to review your manuscript

Author Response

Reviewers' Comments to Author and answers:

To the reviewers,

Thank you both for your very thorough work with our manuscript. We have tried to answer all comments on our best behalf. We submitted a copy with tracked changes.

kind regards

Reviewer 2

The indiscriminate use of antimicrobials to treat calves manifesting diarrhea has been a global concern. However, there are few surveys measuring the degree of responsibility of producers and technicians. The aim of this study was to describe the antibiotic treatment of neonatal calf diarrhea in the four different European countries Austria, Belgium, Portugal and Scotland as part of the United Kingdom using an online survey. The results presented showed that most participants have used antimicrobials to treat diarrhea, and there is some evidence of excessive use of antibiotics and HPCIA for the treatment of NCD. This kind of data is important to highlight the subject, and create demand for specific guidelines and targeted training for veterinarians and farmers.

Despite the highlighted importance of this research, there are several major issues to be addressed by the authors.

Major points:

The authors indicate the restricted use of antimicrobials only for bacterial gastrointestinal infections, however there are limited on-farm tests to detect diarrhea pathogens, besides of high costs. Moreover, the dysbiosis process in the gut, independently of pathogens, can justify the use of antimicrobials in some clinical situations, as recommended by Constable. So, I think the authors should explain this consequence of diarrhea in their manuscript. In my clinical experience, most calves with Crypto diarrhea develop septicemia due to intestinal dysbiosis. Microbiome studies have shown that diarrhea can be a consequence or cause of gut microbiome dysbiosis. So, in my opinion the clinical examination is more important than the identification of infectious agents in the fecal samples. It can help, however, a lot of clinical evidence should be considered, as the authors mentioned in the L116-L135. I recommended an introduction and discussion revision explaining these considerations.

  • Line 77 – 80: The sentence was rewritten to make it more clear that the use of antibiotics in calves with NCD can only be justified in calves suffering from septicemia and bacteremia. In lines 116 ff (Discussion) we address the clinical findings with septicemia in detail.
  • Line 82- 92: Added as requested: Antibiotics may have an impact on the microbiome in the gastrointestinal tract. There are significant differences in microbial diversity between healthy and diarrheic calves within a farm [22,23]. Such microbiome changes in sick calves usually return to pre-diarrheal stage after a week [24]. It is uncertain if the reduction of microbial diversity occurs due to the disease itself or the antibiotic treatment [22]. A very limited number of studies show, that therapeutic antibiotics delay the temporal development of diversity [25]. As example, the use of tulathromycin for treatment appeared to have negative impact on the richness and diversity of the gut microbiome [26]. A study in 2009 showed that calves treated with antibiotics or fed with medicated milk replacer had 70 and 31 % more days with diarrhea, respectively, compared to calves with NCD that only received antibiotics in cases with fever and depression [27].
  • In our clinical experience calves with Crypto rarely become septicemic if they are kept well hydrated and offered sufficient amounts of milk. But this may depend on a number of factors of which we have no scientific evidence.

Only a 1/4 % of participants answered the specific part of the questionnaire that specified which factors influencing decision-making concerning the use of antibiotics. So, the most of participants maybe did not is the person responsible to make the antimicrobial decision in the farm. What is the author interpretation for this finding? please, include this discussion in the text

The design of this questionnaire allowed participants to refuse some of the questions. Therefore, only 287 participants answered the specified questions.  We cannot be sure that every farmer who answered this survey was the one who makes the decisions on their farm. Unfortunately, there is no way to interpret this in any way because we have no data on who the questions where not answered (technical problems, questionnaire too long, questions too complicated, people did not know the answer and gave up…).

It is very nice to present the regulatory rules to the use of antimicrobials in Europe, however what is the experience of authors regarding the effectiveness of penicillin? sulfa and trimetropin? amoxicillin? to treat diarrhea. In my experience the effectiveness is very poor. Show me literature and data regarding this indication. In real life, the indication presented in this paper is effectiveness?

  • Unfortunately, there is no data on pharmacological effectiveness of antibiotic drugs in calves with NCD. In line 29 we emphasize that “there is an urgent need for more scientific evidence to define best practice for treatment of neonatal calf diarrhea”. However, there is indirect evidence of the negative impact, for example Berge et al, 2009 who showed that using antibiotics causes more days of diarrhea. Unfortunately, clinical evidence is the lowest level of scientific evidence and real evidence is missing in this specific topic. On the other hand, Berge and Gomez had good result with algorithms based on clinical signs and treatment concepts for farms.

There are some indications to use Ceftiofur to treat calves manifesting diarrhea and sepsis symptoms. See: textbook of the diseases of cattle, horses, sheep, pigs and goats - two-volume set. Authors: Peter Constable et al.

  • The use of an antibiotic which is effective against an coli will be necessary when it comes to an E. coli induced septicemia. As stated in line: 77- 80.

How to balance the regulatory instruction with the in vivo treatment response?

  • Please explain what you mean by this statement?

Minor points:

L17-L77 - Add biliary excretion to achieve the intestinal lumen

  • We are not 100 % sure what you mean with your statement. We added the sentence “Therefore, the antibiotic must be excreted in bile and reach a sufficient dose in the gastrointestinal tract [21].” in line 81.

L88 due to E.coli K99

  • Line 101: Added as requested.

Add the questionnaire as a supplement document. The link in the M & M is not enough, because is necessary to fill it to follow for the next page

  • We are not sure we understood your remark correctly. In line 260, there is a link presented to see the whole questionnaire online (the link is still active). As this is the very first part of a large study, we do not want to show the whole questionnaire in this manuscript because we intend to publish the rest of the study in another manuscript.

Review the numeration of tables and figures

  • Line 281: Table 2 changed in Table 3.

Tables and figures is not self-explanatory

  • Line 1: Changed as requested: Figure 1. CART: Association of different factors on the use of antibiotics for the treatment of NCD.
  • Line 79: Changed as requested: Table 2. Number of participants using oral and injectable antibiotics by occupation and experience.
  • Line 92: Figure 4. CART: Association of different factors on the use of HPCIA for the treatment of NCD.
  • Material: Figure A1. CART: Association of different factors on the use of antibiotics according to the question “If you usually use antimicrobials please state when”.

I did not found figure A1 (cited at L16 page 6)

  • Line 15 A1 is changed into S1 (Figure S1 is part of the supplementary material which can be found in a separate file): Figure S1. CART: Association of different factors on the use of antibiotics according to the question “If you usually use antimicrobials please state when”.

Table 2 page 8 - absolute numbers or frequencies? it could be performed chi-square

  • Line 79: We changed the caption of Table 2. Number of participants using of oral and injectable antibiotics by occupation and experience.
  • A chi square test cannot be performed with this data because the categories are not mutually exclusive. People can choose to give oral and injectable antibiotics. We simplified the table and deleted two columns to make it easier to read.

Thanks for the opportunity to review your manuscript

Thank you for the comments and the helpful feedback.